# Anthelmintic efficacy of *Holarrhena pubescens* against *Raillietina* spp. of domestic fowl through ultrastructural, histochemical, biochemical and GLCM analysis

**Rachita Saha, Manjil Gupta, Rima Majumdar[ID], Subrata Saha[ID], Pradip Kumar Kar[ID]** *

Parasitology Laboratory, Department of Zoology, Cooch Behar Panchanan Barma University, Cooch Behar, West Bengal, India

* karpradip@gmail.com

**Data Availability Statement:** All relevant data are within the paper and its Supporting information files.

## Abstract

Globally, traditional knowledge systems are a powerhouse of information which can revolutionise the world, if decoded accurately and logically. Plant-based ethno-traditional and folklore curatives/medicines has a firm basis in the psyche of the common masses of West Bengal and *Holarrhena pubescens* is a representative example of it. This article communication on depicting the anthelmintic efficacy of ethanolic extract and Ethyl acetate fraction of the stem bark of *Holarrhena pubescens* against the cestode *Raillietina* spp. through efficacy studies, ultra-structural observations, histochemical and biochemical analysis on some tegumental enzymes i.e., Acid Phosphatase (AcPase), Alkaline Phosphatase (AlkPase), Adenosine Triphosphatase (ATPase) and 5-Nucleotidase (5-Nu) along with Gray Level Co-occurrence Matrix (GLCM) analysis of histochemical study. Praziquantel was used as the reference drug. Investigations revealed 10mg/ml dosage of crude extract was the most efficacious dose and amongst the fractions the ethyl acetate fraction showed the most anthelmintic property. Ultrastructural studies through Scanning Electron Microscope (SEM) and Transmission Electron Microscope (TEM) clearly depicted the damage in head, sucker, proglottids, proximal and distal cytoplasm (DC), microtriches (MT), basal lamina (BL), nuclear membrane (NM), and, nucleolus (NL) in the treated worms. Histochemical studies revealed decrease in staining intensity for all the tegumental enzymes in the treated worms compared to control. The GLCM analysis strongly supported the result of histochemical studies. Biochemical studies revealed marked reduction in enzyme activity in the treated worms with maximum reduction in the activity of 5- Nu (77.8%) followed by ATPase (63.17%).

## Introduction

Neglected Tropical diseases (NTDs) appear globally in about 149 different nations [1], particularly in the less developed ones where socioeconomic status are severely compromised. NTDs —like helminthiasis—cause a greater global burden of ailments rather than more well-known illnesses like malaria and TB, because they are the most prevalent infectious agents affecting humans and animals in underdeveloped nations [2]. Currently, it is estimated that 33% of

**Funding:** The authors received no specific funding for this work.

**Competing interests:** The authors have declared that no competing interests exist.

people living in the developing countries in Sub-Saharan Africa, Asia, and the Americas are infected with helminths [3].

Domestic poultry is one of the major primary source of dietary protein in the world, providing both meat and eggs [4]. Though the productivity of the poultry industry has steadily increased over the years [5], still, the industry faces serious hurdles in production due to helminth infections caused by nematodes, cestodes and trematodes [6–8]. Globally, 66% of the helminth infections are caused by cestodes with *Raillietina* spp. being the most ubiquitous genus in majority of the prevalence studies [9]. The genus *Raillietina* consists of tapeworms belonging to Order Cyclophyllidea and Family Davaineidae. The different species namely *R. echinobothrida*, *R. tetragona* and *R. cesticillus* are identified using light microscopy and scanning electron microscopy by morphometric analysis of the scolex and proglottids position of the genital pore, number of eggs per egg capsule, and rostellar opening surface structures [10]. Their life cycle is completed in two hosts with avian and rodent species being the definitive host and ants and beetles being the intermediate host. The adults live in the duodenum of the host where they get anchored to the intestinal wall, draw nutrition, grow and reproduce [11, 12]. The pathogenicity of *Raillietina* infections in poultry birds is characterised by disruption of intestinal villi, degeneration and necrosis of intestinal epithelial cells, parasitic granulomas, enteritis, anemia, weight loss and decreased egg production [13, 14]. There are few reports of accidental infection of *Raillietina* spp. in humans occurring due to accidental ingestion of cysticercoid infected ants or beetles [15–17].

Widespread use of over-the-counter anthelmintics has led to the development of resistance to these drugs in most animals [18]. Anthelmintic resistance is also an emerging problem in poultry warranting the search for plant-based alternatives. Various phytochemicals namely phenols, alkaloids, saponins, tannins, terpenoids and flavonoids derived from different plants have been shown to act as anthelmintics by altering the cell membrane permeability, interfering with mitochondrial function and energy generation, affecting calcium utilisation, inhibiting neurotransmission with subsequent paralysis and affecting egg hatching [19]. In many underdeveloped nations, livestock producers sometimes lack to access basic veterinary service and medications and medicinal plants continue to be utilised in concoction with veterinary pharmaceuticals to cure animal maladies [20, 21]. So, traditional, indigenous or ethnomedicinal knowledge of ethnic/local population can provide us with valuable inputs for the development of curatives from natural products with anthelmintic potential.

*Holarrhena pubescens* is a medicinal plant used by local Rajbangshi or Koch community in Cooch Behar, West Bengal, India [22, 23]. *Holarrhena pubescens* Wall. ex G. Don, Syn -*Holarrhena. antidysenterica* (Roth) Wall. ex A. DC. belongs to family Apocynaceae, (Common Name: *Indrajao*, *Kutraj*, *Kutaja*, Bitter Oleander, *Kurchi*) and is indigenous to the eastern part of Tropical Africa and Tropical Asia (Indo-China). It is distributed in the subtropical forests of India, Myanmar, Nepal, Bhutan, China and Malaysia [24]. *H. pubescens* is a huge deciduous tree that grows to 9–18 meters in height and is commonly found in plains and mountains [25]. The leaves are oblong and elliptic in shape. Flowers are fragrant white corymbose cymes. The corolla is lobed and oblong in shape. Fruits are long, having thin follicles with white markings. The seeds are globous and linear-oblong in shape. The flowers bloom from April to July and fruiting occurs from August to October [26]. Astringent, antimalarial, antidysenteric, anthelmintic, stomachic, febrifugal, antibacterial, and antioxidant characteristics have been observed in several portions of the plant (stem, root, leaves, and flower) [27]. The methanolic extract of the stem bark of this plant showed hypoglycaemic action [25]. Previous studies has shown that the plant's leaf, bark, fruit, and seed all have therapeutic benefit. The leaf of the plant is employed in pest control, and the entire plant is utilized as green manure. Its applications has been documented in Ayurveda, Homoeopathy, and folklore medicine. The flavonoids and phenolic components in *Holarrhena pubescens* aqueous and methanolic bark extract showed

high antioxidant activity [25] *H. pubescens* alkaloids were shown to be useful in treating acute dysentery, inflammatory bowel illness, diarrhoea, and, worm infections in the gut [28]. It is also reported that *H. pubescens*, may be helpful to an extent, in treating Amoebiasis [29] and this plant also has anti-diabetic, anti-amnesic and anti-inflammatory property [28].

About 40 alkaloids have been found in the stem bark, root bark, and seeds of *Holarrhena pubescens* [24]. The barks of stem and root (in total) had the greatest concentration of alkaloids (up to 4.3%), followed by leaves (1.0–1.5%) and seeds (0.6–1.8%). Only 0.4% of the 4.3% alkaloids present in the barks of stem and root are accessible to blooming plants. According to the Ayurvedic Pharmacopoeia of India [30], the bark contains around 2% alkaloids. This plant's stem bark, seeds, and roots were found to be high in steroidal alkaloid compounds such asiso-conessine, kurchine, conessidine, conkurchicine, and holarrhimine, as well as conessine, which possesses antimalarial and antibacterial activities [31, 32].

Whole worm-based assays are commonly used for identification of new anthelmintic compounds. Unlike target based approaches, the main focus of these assays are whether the compounds are able to kill or disable the worms *in vitro* [33]. Worm motility, morphological changes, ATP production and enzyme activity are some of the features assessed in these assays [34]. The current investigation focuses on the plausible anthelmintic property of the ethanolic extract of stem bark of *H. pubescens* through ultrastructural, histochemical, GLCM analysis and biochemical studies of the alterations in the activity of the tegumental enzymes namely Acid Phosphatase (AcPase), Alkaline Phosphatase (AlkPase), Adenosine Triphosphatase (ATPase) and 5-Nucleotidase (5-Nu) of *Raillietina* spp *in vitro*.

## Materials and methods

### Preparation of extract from *Holarrhena pubescens*

Fresh bark was collected from the plant *H. pubescens* from the town of Cooch Behar (26.3169˚ N, 89.4457˚ E), West Bengal, India in month of April- June, Since 2018 till date. Dr. Monoranjan Chowdhury identified the plant and the accession number is- 12638. It was kept at herbarium of Department of Botany, University of North Bengal, Darjeeling, West Bengal. The plant materials were finely chopped and air dried for 24–36 hours after being collected. The dried bark components were soaked in ethanol in a glass container (100 grams in 500 ml) for 15–20 days with frequent stirring. The solution was then filtered using Whatman Filter Paper (No. 14), and then dried using a Rotary Vacuum Evaporator (Buchi Rotavapor R-100). After drying, the collected crude extracts of plant material were stored under refrigeration at 4˚C until further use. Around 7.0–8.0 gram (g) of plant extract was obtained from 100 g of plant material immersed in 500 millilitre (ml) ethanol. Different fractions of the obtained ethanolic extract were separated using solvents with varying polarities such as Hexane, Chloroform, Ethyl acetate, and *n*-Butanol using a separating funnel and fractional distillation process [35]. 50 ml aqueous solution of crude extract (2.5 gm in 50 ml distilled water) was combined with 50 ml of Hexane and left to stand for 2 hours. The Hexane fraction was subsequently removed from the upper phase using the liquid liquid separation technique. The method was repeated three times using Chloroform (lower phase), Ethyl acetate, and n-Butanol. The final volume of each fraction (150 ml) was collected and dried using a Rotary Vacuum Evaporator (Buchi Rotavapor R-100). Around 0.25 g, 0.14 g, 0.92 g and 1.1 g of Hexane, Chloroform, Ethyl acetate and *n*-Butanol fractions were obtained from 2.5 g of crude extract respectively.

### Collection of parasites

Fresh intestines of *Gallus gallus domesticus* were obtained from a local market, and live parasites (*Raillietina* spp.) were isolated from the gut in 0.9 percent Phosphate-Buffered Saline

(PBS: NaCl-8g, $KH_2PO_4$-0.34g, and $KHPO_4$-1.21g in 1 litre (l) distilled water, pH 7.4) and kept in an incubator set at 37±1˚ C.

## Experiments

**Efficacy testing.** The model parasite i.e., *Raillietina* spp. were incubated in glass petridish at 37˚C in PBS without plant extract (control), and with Ethanolic crude extract at 1.0, 2.0, 5.0 and 10.0 mg/ml concentration respectively. Praziquantel was used as the reference drug. Three replicates were used for each concentration. The time required for complete inactiveness or paralysis and death of the parasites were recorded. The time for paralysis was determined by observation of the parasites after treatment. Paralysis was considered when the parasites showed loss of smooth undulating motion and death was considered when there was complete absence of any motion in the parasites. Death was further confirmed if the parasites showed no motion even after transferring to warm PBS. After determining the most efficacious dose of the crude extract, efficacy tests were performed with the same dose of different fractions.

**Scanning Electron Microscopy.** For Scanning Electron Microscopy (SEM) the parasites were fixed in 10% Neutral Buffered Formalin (NBF) at 4˚C for 4 hours followed by repeated washing in double-distilled water, dehydration through acetone grades, critical-point-drying (CPD) using liquid $CO_2$, metal-coating with gold palladium, and viewed in a JEOL-JSM-35 CF scanning electron microscope [36].

**Transmission electron microscopy.** Control and treated worms were fixed in Karnovosky's fixative (4% Paraformaldehyde and 1% Glutaraldehyde) in 0.1M Sodium Phosphate buffer (pH 7.4) and then processed for Transmission Electron Microscope TECNAI G20 HR-TEM as per usual method [37] for ultrastructural studies [36].

**Histochemical studies.** Following Pearse 1968 [38], histochemical localisation of Acid Phosphatase (AcPase), Alkaline Phosphatase (AlkPase), and Adenosine Triphosphatase (ATPase) in control and phytoproduct exposed worms was carried out [39]. The Wachstein and Meisel 1957 [40] approach, which uses adenosine monophosphate as a substrate, was used to examine 5- Nucleotidase (5-Nu). For 30 minutes, sections were incubated at 37˚C in a freshly produced medium containing 10 ml of 1.25 percent adenosine-5-phosphate, 5 ml of 0.2 M Tris buffer at pH 7.2, 30 ml of 0.2 percent $Pb(NO_3)_2$, and 5 ml of 0.1 M $MgSO_4$. All parasites were cut in 12μm-14μm thickness through Cryostat (Leica- CM 3050S) at -15˚C -18˚C [41].

**GLCM analysis.** Gray Level Co-occurrence Matrix (GLCM) analysis for the quantification of the histochemical localization of the enzymes was performed following Dragić *et al.* 2019 [42]. Mean Gray Value (MGV) and Integrated Density (IG) were calculated for all the images. Five parameters namely Angular Second Moment (ASM), Contrast (CON), Co-relation (COR), Inverse Different Moment (INV) and Entropy (ENT) were calculated for GLCM analysis. Out of these, ASM, COR and INV measures the homogeneity of the image and should be more in deeply stained images whereas CON and ENT measures contrast and lack of spatial organization in the image and should be less in deeply stained images. Images obtained from histochemical study for different enzymes were cropped for region of interest (comprising of the tegument, sub tegument and parenchyma), converted to grayscale and subjected to GLCM analysis through the software Image J [43]. A set of 20 images each from different categories like control, Praziquantel treated, crude extract treated and Ethyl acetate fraction treated was used for GLCM analysis for each enzyme. Receiver Operating Characteristic (ROC) curve analysis was also performed to validate the results obtained from GLCM analysis.

**Biochemical studies.**   The method of Plummer, 1988 [44], was used to estimate the activities of AcPase and AlkPase [36]. ATPase activity was determined by estimating the free phosphate released using the method of Kaplan, 1957 [45], and 5' -Nu was determined by the Bunitian method following Giri and Roy, 2014 [39]. Protein content was calculated using Lowry's method [46] for all the enzyme assays. The total enzyme activity was determined for AcPase and AlkPase as the amount in micromole (μM) of p-nitrophenol formed per minute per gram wet tissue. In case of ATPase and 5- Nu, activity was expressed by liberation of μM inorganic phosphate group per mg protein. The activity of tegumental enzymes was expressed by total and specific enzyme activity.

# Results

## Efficacy testing

The results indicate that the Ethanolic crude extract of *Holarrhena pubescens* and the fractions of ethanolic extract showed anthelmintic activity on cestode worm *Raillietina spp*. The control worms survived in PBS for around 71 hours. In the presence of varying concentrations of the ethanolic extract and different fractions, it was observed that the survival time of the treated worms was longer than that of those treated with Praziquantel (PZ) but lower than the Control (C) (Table 1, Fig 1). The concentration of 10 mg/ml was determined as the most efficacious dose and used for further studies. Out of the different fractions used, the Ethyl acetate fraction (EAF) showed most anthelmintic activity.

## SEM observations

In the control worms, the entire surface of the scolex and proglottids shows a smooth appearance due to being covered with dense microtriches (Fig 2a, 2e and 2i). Suckers are smooth with distinct hooks. In the worms treated with the plant material, distortions are visible in the head region characterised by folding of the tegument, eversion of the suckers and loss of the hooks encircling them. In the proglottids, the tegument shows a folded and cracked appearance (Fig 2c, 2d, 2g, 2h, 2k and 2l). Similar observations were also found in the Praziquantel-treated worms (Fig 2b, 2f and 2j). The damage is more apparent in the worms treated with Praziquantel and Ethyl acetate fraction.

**Table 1. Results of efficacy testing of the ethanolic crude extract of *Holarrhena pubescens* and fractions of the ethanolic extract on *Raillietina* spp.**

| Incubation Medium | Dose (mg/ml) PBS | Time of paralysis (hours) | Time of death (hours) |
|---|---|---|---|
| Control | - | - | 71.0±0.59 |
| Praziquantel | 0.05 | 0.49±0.05 | 0.98±0.04* |
| Ethanolic crude extract | 1.0 | 9.81±0.47 | 12.84±0.48* |
| | 2.0 | 5.09±0.15 | 8.52±0.16* |
| | 5.0 | 2.11±0.23 | 6.89±0.36* |
| | 10.0 | 1.02±0.13 | 4.44±0.21* |
| *n*-Hexane fraction | 10.0 | 6.16±0.35 | 20.23±0.31* |
| Chloroform fraction | 10.0 | 5.32 ± 0.28 | 19.44±0.25* |
| Ethyl acetate fraction | 10.0 | 0.83 ± 0.21 | 3.72±0.52* |
| *n*-Butanol fraction | 10.0 | 1.08 ± 0.33 | 4.66±0.52* |

*Data represents mean value ±SD.

n = 450. Values significant at P<0.05

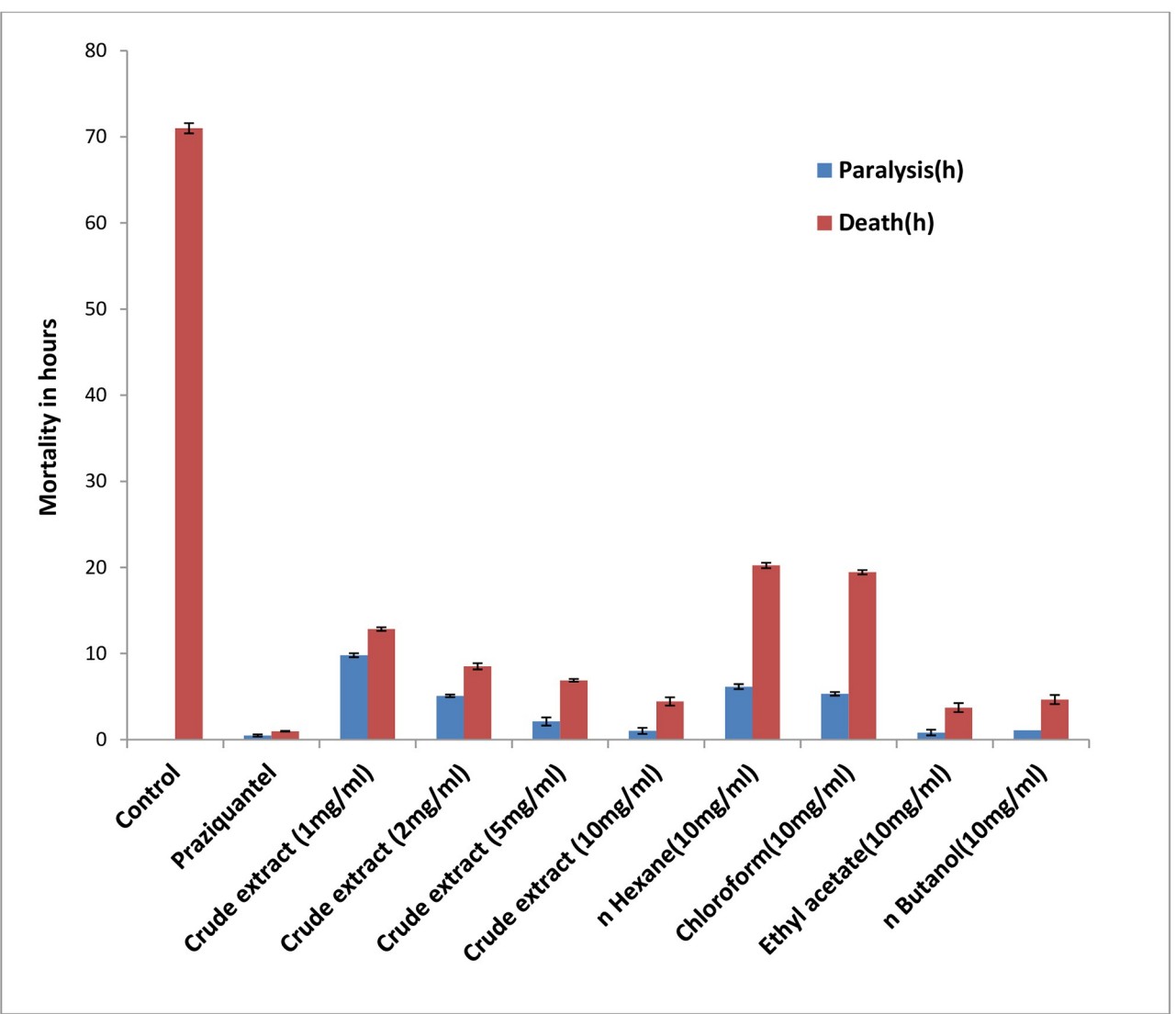

**Fig 1. Graph showing the paralysis time and death time of *Raillietina* spp treated with different doses and fractions of the ethanolic extract of *Holarrhena pubescens*.**

## TEM observations

The control worm's surface, proximal and distal cytoplasm (DC) with microtriches (MT), basal lamina (BL) (Fig 3a) nucleus with double nuclear membrane (NM), nucleolus (NL), chromatin granules (Fig 4a and 4e), and mitochondrial characteristics were all normal, according to transmission electron microscopic studies. Parasites treated with Praziquantel had severe damage in the microtrix layer and distal cytoplasm (Fig 3b). Swollen nuclei with an anomalous singlet nuclear membrane (Fig 4b) and deformed mitochondria lacking cristae were also observed (Fig 4f). The distal cytoplasm of the worms exposed to the plant extract (crude) was completely eroded, and, the basal lamina of the microtriches was exposed to the outer surface (Fig 3c). The nucleolus disintegrated and a swollen nucleus with an uneven nuclear membrane was observed (Fig 4c). In the parasites treated with the plant extract, the

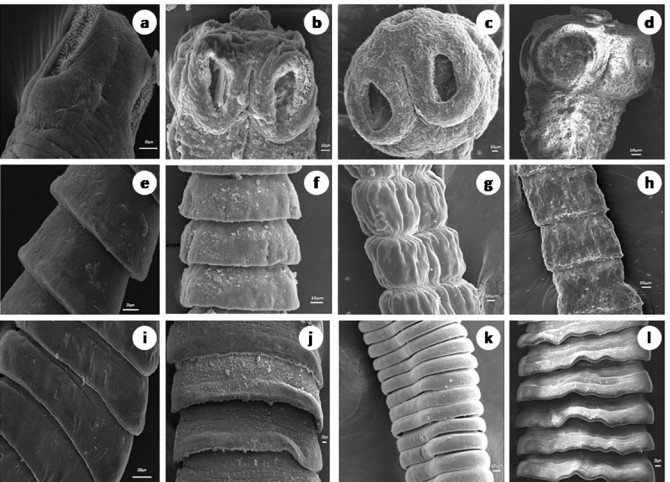

**Fig 2.** Scanning electron micrographs of the head (a—d), neck (e—h) and proglottids (i—l) of control and treated *Raillietina* spp. Control (a, e, i), Praziquantel exposed (b, f, j), Ethanolic crude extract exposed (c, g, k), Ethyl acetate fraction exposed (d, h, l) (all scale bars = 20µm).

majority of the mitochondria developed deformities (Fig 4g). *Raillietina* spp., treated with the Ethyl acetate fraction, similarly exhibited fully damaged microtriches and distal cytoplasm (Fig 3d), rupture of the nuclear membrane, condensation of chromatin granules (Fig 4d) and disintegration and vacuolization of the mitochondrial membrane (Fig 4h).

## Histochemical observation

The tegument of control *Raillietina* spp. showed intense activity of AcPase, AlkPase, ATPase and 5' Nu compared to sub-tegument, somatic musculature, and parenchyma (Figs 5a–5h and 6a–6h). ATPase activity was almost negligible in the tegument and sub-tegument of the parasite treated with crude and Ethyl acetate fraction of the plant extract compared to control (Fig 6c and 6d). A reduced activity of AlkPase was observed throughout the treated sections of the parasite (Fig 5g and 5h). In the control sections of histochemical localization of AcPase, pronounced stain intensity was observed in the tegument and sub- tegument regions (Fig 5a–5d). However, there was minimal AcPase activity oberved throughout the section of parasite exposed to crude, Ethyl acetate fraction and Praziquantel. The 5' -Nu activity was also found to be reduced throughout the tegumental and sub-tegumental region in the phytoproducts exposed worms compared to the control (Fig 6e–6h).

## GLCM results

The calculated values of different GLCM parameters are depicted in Table 2. For ASM, COR and INV, the values are more in the control worms than in the treated worms and for MGV, ID, CON and ENT the control worms show lower values than the treated worms. The results conform to the expected results. MGV and ID showed the lowest value in the control worms of AcPase due to darker staining section compared to the other enzymes. ASM and CON showed highest and lowest values respectively in the control worms for ATPase. COR shows highest vale in the control worms of AcPase. INV has highest value in the control worms of ATPase. ENT has lowest value in the control worms of ATPase. The ROC curve analysis is

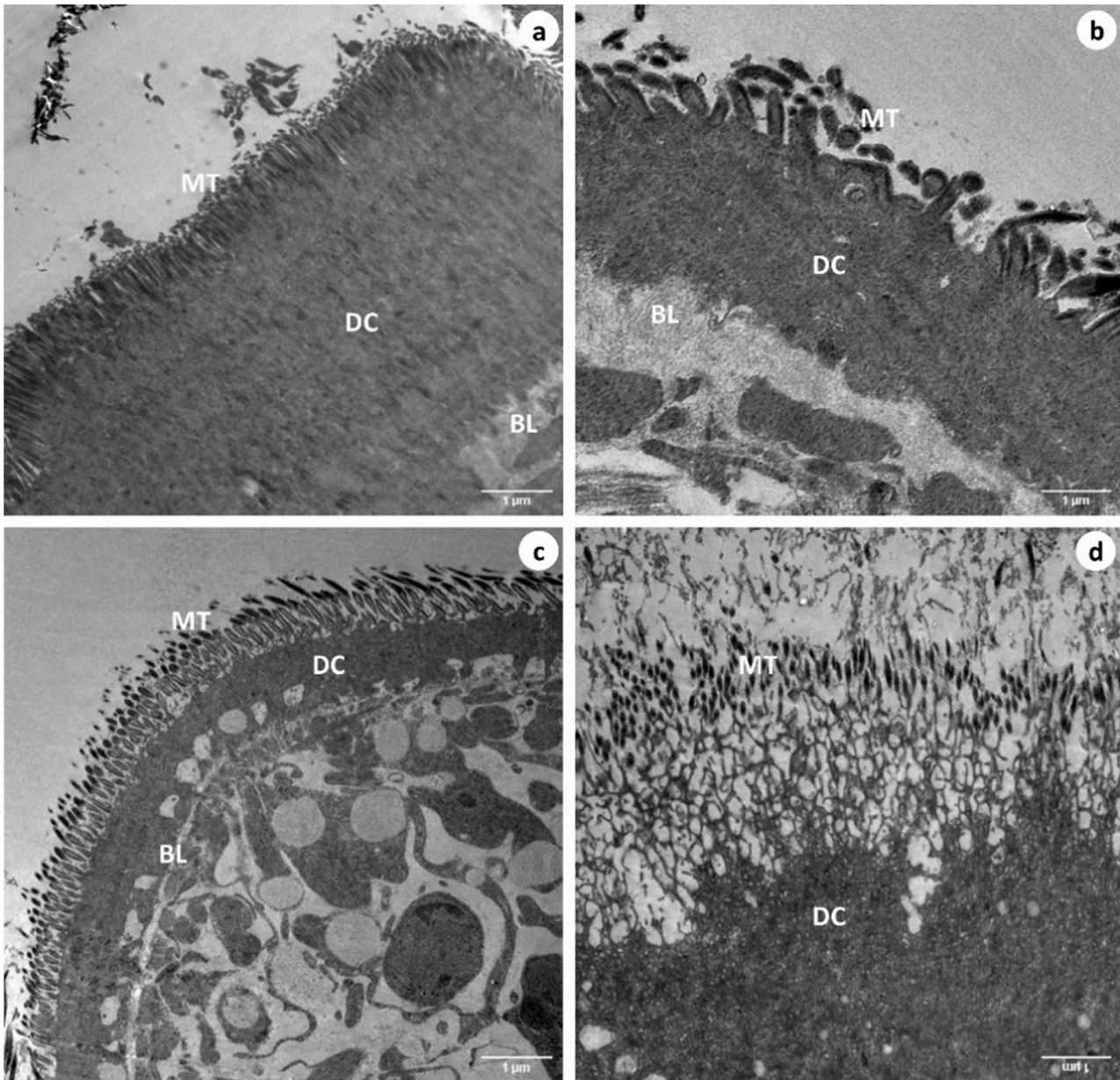

**Fig 3.** Transmission electron microscopy images of sections of Control (a), Praziquantel (b), Ethanolic crude extract (c), Ethyl acetate fraction treated (d) *Raillietina* spp. showing Microtrix layer (MT), distal cytoplasm (DC) and basal lamina (BL) (all scale bar = 1μm).

shown in Fig 7. The area under the curve (AUC) values is depicted in Table 3. All the values are greater than 50% indicating that the process is considerable.

## Biochemical observation

The quantitative analysis of the tegumental enzymes of *Raillietina* spp. showed significant impact on treatment with the plant materials and the reference drug Praziquantel. As shown in results (Table 4 and Fig 8) AcPase activities decreased by 50.33, 39.70 and 60.56 percent, AlkPase activities decreased by 54.05, 65.44 and 39.74 percent, ATPase activities decreased by

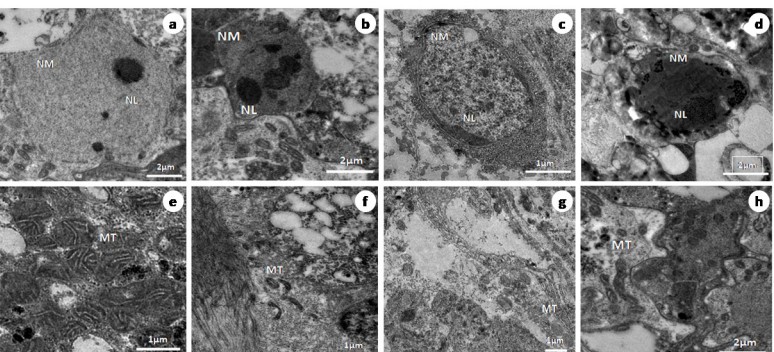

**Fig 4.** Transmission electron microscopy images of sections of Control (a, e), Praziquantel treated (b, f), Ethanolic crude extract (c, g) Ethyl acetate fraction (d, h) treated *Raillietina* spp showing nucleus (N), nucleolus (NL) and nuclear membrane (NM) and mitochondria (M); Scale bars = 0.5 μm (a—d) and 0.2 μm (e—h).

37.16, 41.38 and 63.17 percent, and 5´-Nu activity decreased by 71.63, 65.75, and 77.80 percent, after exposure to the Praziquantel, Ethanolic crude extract, Ethyl acetate fraction of the crude extract respectively. In the worms fed with the Ethyl acetate fraction of the plant, the highest inhibition of activity was observed in the case of 5´- Nu followed by ATPase.

## Discussion

In our present study, the ethanolic crude extract treated worms showed a dose dependent increase in the time of paralysis and death indicating the putative anthelmintic potential of *Holarrhena pubescens*. The control worms survived for a much longer period than the worms treated with either praziquantel or plant extracts under similar conditions. Amongst the fractions of the ethanolic extract, the worms treated with the Ethyl acetate fraction showed the highest reduction in motility and survivability.

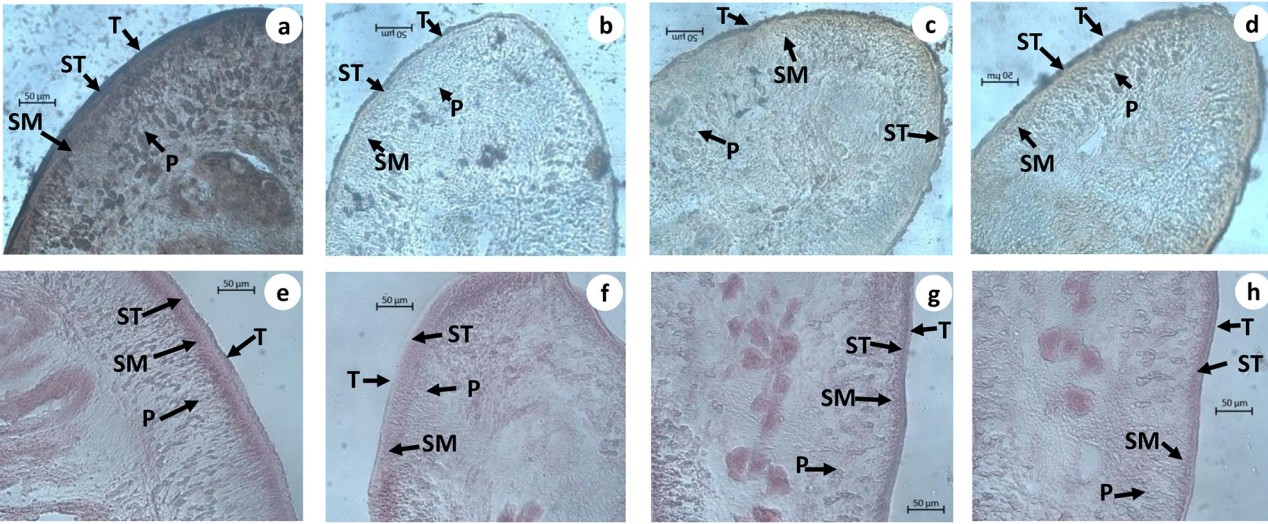

**Fig 5.** Histochemical evidence of AcPase (a—d) and AlkPase (e—h) activity in *Raillietina* spp., T-Tegument, ST -Subtegument, SM-Somatic musculature, P-Parenchyma, (a, e). Control, (b, f) Praziquantel, (c, g) Crude extract (d, h) Ethyl acetate fraction treated (All scale bars = 50 μm).

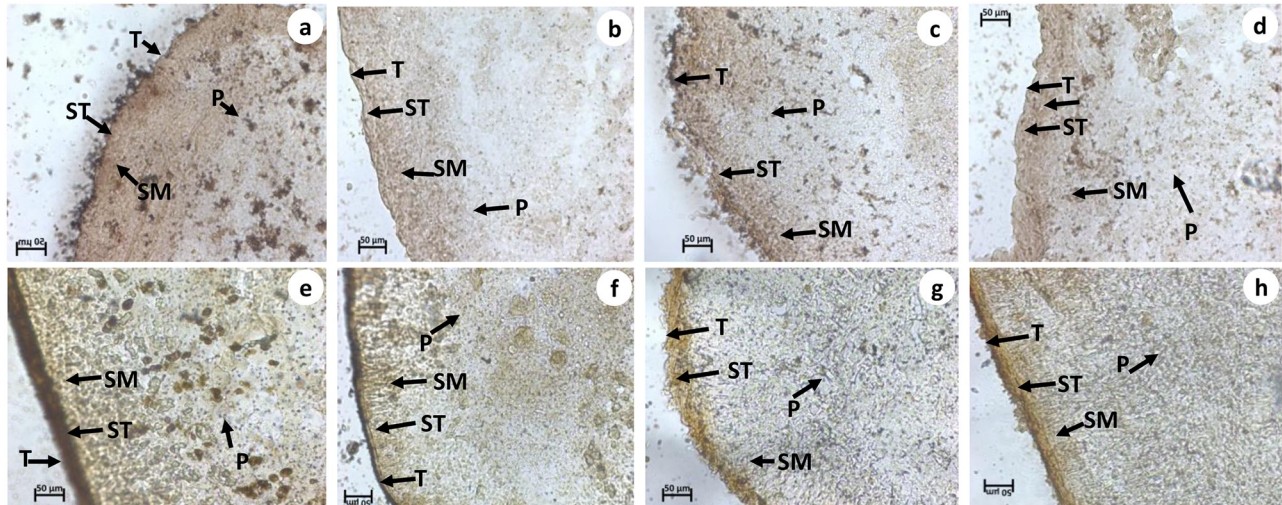

**Fig 6.** Histochemical evidence of ATPase (a—d) and 5'-Nu (e—h) activity in *Raillietina* spp., T-Tegument, ST -Subtegument, SM-Somatic musculature, P-Parenchyma, (a, e) Control, (b, f) Praziquantel, (c, g) Crude extract treated, (d & h) Ethyl acetate fraction treated (All scale bars = 50 μm).

The tegument in cestodes forms a digestive-absorptive-protective surface at the host para-site interface [47, 48]. Ultrastructural studies revealed the presence of polymorphic micro-triches in the entire region of the scolex as well as in, immature and mature proglottids, suggesting their role in the absorption of nutrients through the whole body surface [49]. The action of most anthelmintic drugs depend on their passive transfer through this external

**Table 2. GLCM parameters of images obtained from histochemical studies of different enzymes in *Raillietina* spp.**

|  |  | MGV | ID | ASM | CON | COR | INV | ENT |
|---|---|---|---|---|---|---|---|---|
| AcPase | C | 113.03 ±15.07 | 7064539.0 ±941926.3 | 0.000473 ±0.000112 | 189.95 ±58.36 | 0.0022 ±0.0006 | 0.13 ±0.0171 | 7.99 ±0.26 |
|  | PZ | *168.67 ±16.03 | *10542149.4 ±1002093.42 | *0.000172 ±0.000027 | *558.7 ±66.56 | *0.00081±0.00009 | *0.07 ±0.0054 | *8.95 ±0.12 |
|  | ECE | *136.82 ±31.29 | *8551189.8 ±1955700.6 | *0.000372 ±0.00015 | *331.26 ±218.26 | 0.00172 ±0.00071 | *0.10 ±0.0285 | 8.28 ±0.5 |
|  | EAF | *146.41 ±34.33 | *9306178.5 ±2190156.44 | *0.000299 ±0.000153 | *433.75 ±239.09 | *0.00137±0.00063 | *0.09 ±0.0296 | *8.53 ±0.54 |
| AlkPase | C | 160.11 ±3.18 | 15287542.5 ±303448.53 | 0.000248 ±0.000011 | 311.32 ±38.36 | 0.001±0.0 | 0.09 ±0.0431 | 8.59 ±0.05 |
|  | PZ | *164.71 ±5.4 | *15726452.2 ±515784.83 | 0.000234 ±0.00004 | *383.33 ±33.5 | 0.001±0.0 | *0.08 ±0.0045 | 8.67 ±0.14 |
|  | ECE | *169.76 ±5.79 | *15930277.4 ±671090.57 | 0.000242 ±0.000027 | *411.49 ±55.56 | 0.00099±0.00002 | *0.08 ±0.0073 | *8.67 ±0.1 |
|  | EAF | *167.65 ±8.07 | *15825408.9 ±715618.67 | 0.000236 ±0.000045 | *445.2 ±56.4 | *0.00099±0.00002 | *0.08±0.01125 | *8.71 ±0.12 |
| ATPase | C | 175.75 ±12.06 | 67556571.1 ±4636486.82 | 0.00079 ±0.000282 | 88.9 ±14.54 | 0.00157 ±0.00053 | 0.23 ±0.0432 | 7.59 ±0.43 |
|  | PZ | 182.64 ±16.28 | 70208272.0 ±6259795.72 | *0.00053 ±0.000124 | *99.49 ±11.8 | 0.00128 ±0.00048 | *0.14 ±0.0089 | *7.96 ±0.18 |
|  | ECE | 184.07 ±9.89 | 70534709.2 ±3908405.44 | *0.00035 ±0.000071 | *124.57 ±15.59 | *0.00107±0.00032 | *0.19±0.0101 | *8.33 ±0.15 |
|  | EAF | 183.12 ±7.26 | 70442003.2 ±2923379.36 | *0.00034 ±0.000066 | *118.6 ±12.49 | *0.001±0.00034 | *0.15 ±0.008 | *8.35 ±0.13 |
| 5'- Nu | C | 169.78 ±8.13 | 69543524.8 ±3329793.72 | 0.00028 ±0.00003 | 153.14 ±26.47 | 0.00098 ±0.00007 | 0.14±0.0127 | 8.53 ±0.1 |
|  | PZ | *19.37 ±6.13 | *77976486.4 ±2509017.94 | *0.0002 ±0.000031 | *173.83 ±29.83 | *0.00054±0.00008 | *0.12 ± 0.0099 | 8.62 ±0.13 |
|  | ECE | *178.98 ±7.15 | *73310255.7 ±2928377.66 | *0.00023 ±0.000022 | *190.3 ±24.62 | *0.00083±0.00006 | *0.12 ±0.0142 | *8.68 ±0.08 |
|  | EAF | *178.65 ±6.34 | *73228978.2 ±2717574.88 | *0.00025 ±0.00004 | *176.1 ±27.34 | *0.0009±0.00009 | *0.12 ±0.0132 | *8.8 ±0.14 |

C- Control, PZ-Praziquantel treated, ECE-Ethanolic Crude Extract treated, EAF- Ethyl acetate fraction treated.

Data represent Mean±SD.

* Values significant at P<0.05

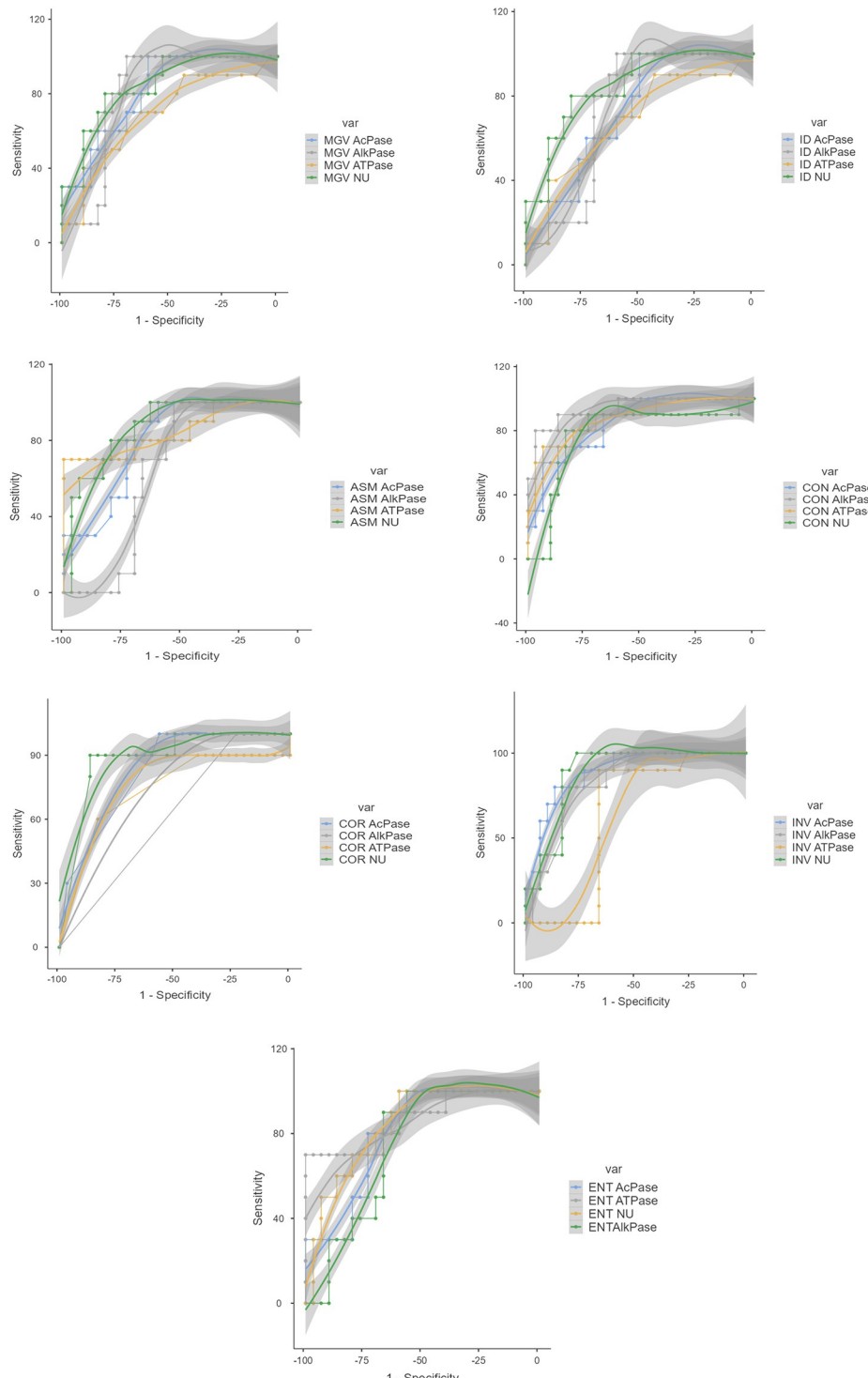

**Fig 7. ROC curves for GLCM parameters of images obtained from histochemical studies on different enzymes in** *Raillietina* **spp.**

**Table 3. Area under curve (AUC) values of GLCM parameters of images obtained from histochemical studies of different enzymes in *Raillietina* spp.**

| Enzyme | Area under curve (AUC) values (%) | | | | | | |
|---|---|---|---|---|---|---|---|
| | MGV | ID | ASM | CON | COR | INV | ENT |
| AcPase | 81.00 | 72.00 | 81.00 | 86.00 | 82.00 | 88.00 | 80.00 |
| AlkPase | 80.00 | 74.00 | 85.00 | 94.00 | 62.00 | 83.00 | 74.00 |
| ATPase | 69.00 | 68.00 | 87.00 | 88.00 | 74.00 | 63.00 | 87.00 |
| 5'- Nu | 84.00 | 84.00 | 65.00 | 78.00 | 88.00 | 88.00 | 84.00 |

Mean Gray Value–MGV, Integrated Density–IG, Angular Second Moment—ASM, Contrast- CON, Co-relation—COR, Inverse Different Moment—INV and Entropy–ENT.

surface i.e. the tegument [50]. The ultrastructural alteration in the tegument are therefore used as a reliable indicator of anthelmintic activity of a compound [51]. Ultrastructural studies (SEM) showed distortion of the smooth texture of the tegument as well as cracking and folding in the treated worms. This is likely to interfere with the nutrient uptake and energy metabolism of the parasite. Similar type of alterations/damages to the tegument by phytoproduct treated worms have been reported by many authors [41, 52–54]. Albendazole also showed similar types of effects on the tegument [55]. The TEM images further show that the microtrich layer, nuclear membrane, chromatin as well as mitochondria, all suffer damages in the phytoproduct treated worms. This indicates the interference of the phytoproducts with the energy metabolism of the worms as well as functioning of the nucleus. Further, the surface topography and subcellular structures are distorted by the plant extract. The findings are consistent with similar studies with different plant extracts [39, 41, 56, 57].

Tegumental enzymes play key role in maintaining ionic homeostatsis, active transport and metabolic regulation within the parasites [58]. In the present study, the tegumental enzymes namely Acid Phosphatase, Alkaline Phosphatase, Adenosine Triphosphatase and 5'-Nucleotidase show pronounced activity in the tegument, subtegument, somatic musculature and parenchyma of the control worms. However, in the phytoproduct as well as Praziquantel treated worms, there is a marked reduction in the activity of all the four enzymes in the aforementioned regions. Similar results were also evident from the biochemical studies wherein there is a marked reduction in the activity of the tegumental enzymes in the Praziquantel and phytoproduct treated worms. Similar type of results were obtained after treatment with other plant extracts as well [39, 59, 60]. Acid Phosphatase is an important lysosomal enzyme in helminths [58] and the reduction in its activity indicates the disruption of lysosomes in the phytoproduct treated worms [61]. Alkaline Phosphatase is essential for the uptake of glucose through the

**Table 4. Results of biochemical analysis of enzyme activity in *Raillietina* spp.**

| | Tegumental enzyme (Total/Specific) activity in μM | | | | Percentage changes after treatment (%) | | | |
|---|---|---|---|---|---|---|---|---|
| | AcPase | AlkPase | ATPase | 5´-Nu | AcPase | AlkPase | ATPase | 5'- Nu |
| C | 21.71± 1.99/9.07 ± 0.87 | 30.09±1.62/11.74±1.25 | 244.27±13.96/45.78±4.07 | 771.15±22.06/159.93±18.5 | - | - | - | - |
| PZ | 11.84±1.39/*4.50±0.65 | 13.97±1.36/*5.40±0.73 | 153.68±13.43/*28.77±2.15 | 234.98±33.01/*45.37±4.7 | 50.33 | 54.05 | 37.16 | 71.63 |
| ECE | 12.98±0.7/*5.47± 0.76 | 21.35±1.74/*8.21±0.99 | 150.25±11.9/*26.84±2.83 | 287.6±30.35/*54.78±7.85 | 39.7 | 65.44 | 41.38 | 65.75 |
| EAF | 8.33±1.09/*3.58± 0.52 | 17.68±1.96 /*7.08±0.89 | 89.54±8.97/*16.86±1.95 | 202.38±24.84/*35.49±5.92 | 60.56 | 39.74 | 63.17 | 77.8 |

C- Control, PZ- Praziquantel treated, ECE- Ethanolic crude extract treated, EAF- Ethyl acetate fraction treated.

Data represents mean value ± SD.

*Values are significant at P ≤ 0.05.

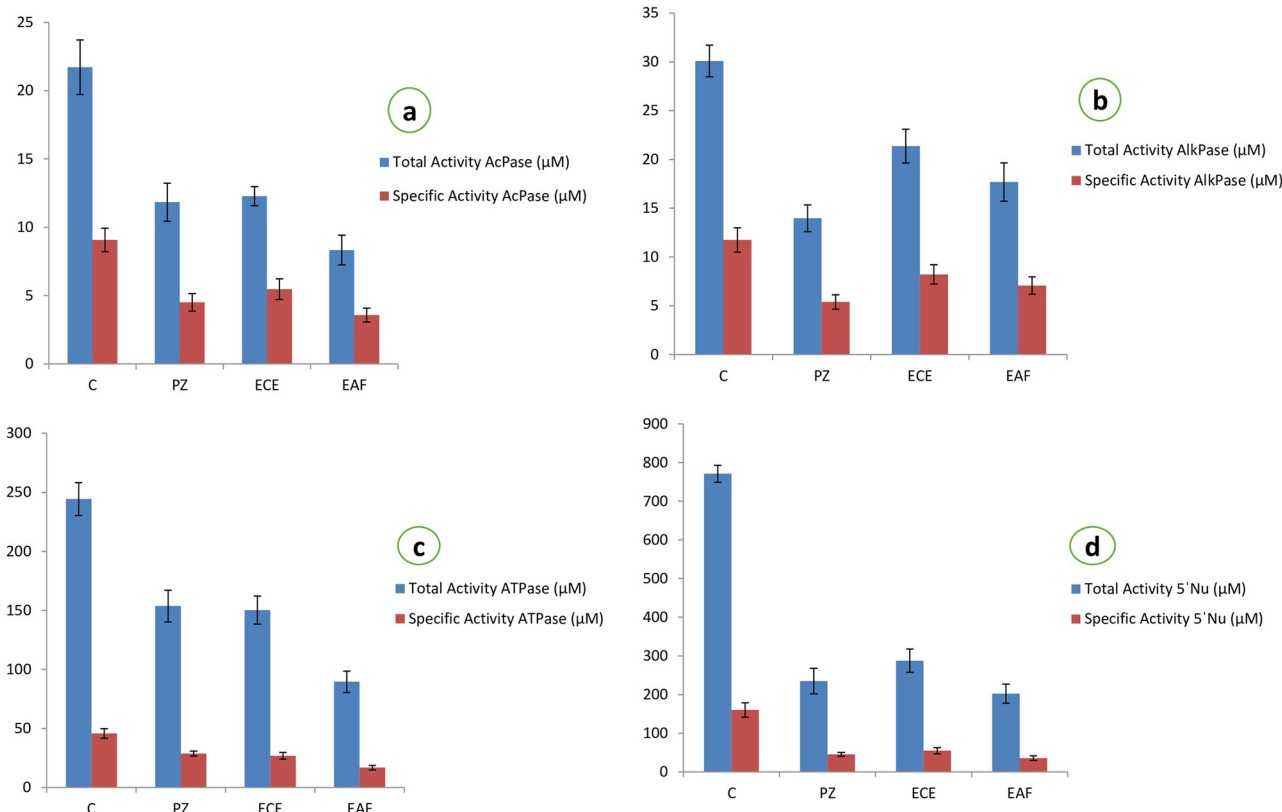

**Fig 8. Graph showing the effect on total and specific activities of different tegumental enzymes of *Raillietina* spp. treated with crude extract and fractions of *Holarrhena pubescens*.** C- Control, PZ- Praziquantel treated, ECE- Ethanolic crude extract treated, EAF- Ethyl acetate fraction treated.

tegument [62] and reduction in its activity may interfere with the glucose uptake as well as energy metabolism in the worms [36, 41]. Adenosine Triphosphatase in helminths plays a role in the transport of various ions and phospholipids across the tegument [63] and reduction in its activity indicates that the crude plant extract as well as the Ethyl acetate fraction interferes with the transport of different ions as well as the energy metabolism of the worms. Reduction in 5'-Nu activity in the treated worms suggest interference with the uptake of nucleosides [64].

Texture analysis of images is used to quantify spatial differences in the pixels. Gray Level Co-occurrence Matrix (GLCM) is one such method of image texture analysis. It has been used for the enzyme histochemistry in neural tissues [65, 66] and analysis of chromatin architecture [67]. We, for the first time are reporting the use of GLCM for enzyme histochemistry analysis in invertebrates. The MGV represents the brightness of the image and the ID is equivalent to the product of the area and MGV. So, the values of these two parameters are lower in the control images as because the intensity of the staining is higher for all the enzymes in the control images compared to the treated images. ASM and INV measure tissue homogeneity and may increase due to the changes in tissue texture due to increased staining. In our analysis, both ASM and INV are higher in the control images with higher staining intensity than in the phytoproduct treated images. CON measures local differences in the image and ENT measures the lack of spatial organisation. Essentially, CON and ENT will increase with the decrease in image homogeneity. So, the values of CON and ENT are lower in the control images compared to the phytoproduct treated images of the worms. High COR values indicate that the pixels are

dependent on one another and vice versa. We obtained slightly higher COR values in control images as compared to treated. From Table 2 it is evident that CON and INV are better indicators of the enzyme histochemistry results.

## Conclusion

The results of efficacy and ultrastructural analysis clearly show that the stem bark Ethanolic crude extract and Ethyl acetate fraction of *Holarrhena pubescens* has anthelmintic potential. Histochemical and biochemical analysis has shown marked reduction in the activity of tegumental enzymes, i.e., AcPase, AlkPase, ATPase and 5´- Nu of the *Raillietina* spp. The result of various experiments has shown that the Ethyl acetate fraction of crude extract of stem bark of *H. pubescens* has higher effective anthelmintic property. Our study firmly establishes the anthelmintic potential of *H. pubescens*. Further studies are required on the identification of the active components of the ethanolic extract as well ethyl acetate fraction of the plant and mechanism of their action, so that, they can be used as candidates for the development of anthelmintic drugs for the control of Raillietinosis in poultry.

## Supporting information

**S1 Table. Results of efficacy testing of ethanolic crude extract of *Holarrhena pubescens* and its fractions against *Raillietina* spp.**
(XLSX)

**S2 Table. Data for biochemical analysis of enzyme activity- Acid Phosphatase and Alkaline Phosphatase.**
(XLSX)

**S3 Table. Data for biochemical analysis of enzyme activity- 5' Nucleotidase and ATPase.**
(XLSX)

**S4 Table.**
(ZIP)

## Acknowledgments

The authors gladly appreciate the constant support and help in conducting the experiments with instrumental and infrastructural facility of the Department of Zoology, Cooch Behar Panchanan Barma University, Cooch Behar. We very much thankful to Department of Botany, University of North Bengal, Darjeeling, West Bengal for the accession number of the plant. The authors are grateful to SAIF, North Eastern Hill University (NEHU), Shillong and Centre for Research in Nanoscience and Nanotechnology (CRNN), Kolkata for their technical support. We also acknowledge the TEM facility of SAIF, All India Institute of Medical Science (AIIMS), New Delhi for their kind cooperation.

## Author Contributions

**Conceptualization:** Rachita Saha, Pradip Kumar Kar.

**Data curation:** Rachita Saha, Manjil Gupta.

**Formal analysis:** Rachita Saha, Manjil Gupta, Rima Majumdar, Subrata Saha.

**Investigation:** Rachita Saha, Manjil Gupta, Rima Majumdar.

**Methodology:** Rachita Saha, Manjil Gupta, Rima Majumdar.

**Software:** Manjil Gupta.

**Supervision:** Pradip Kumar Kar.

**Validation:** Pradip Kumar Kar.

**Writing – original draft:** Rachita Saha, Manjil Gupta.

**Writing – review & editing:** Manjil Gupta, Pradip Kumar Kar.

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
