## [Decision Letter · Decision Letter 0]

6 Mar 2023

PONE-D-23-03097Anthelmintic efficacy of * Holarrhena pubescens * against * Raillietina * spp. in domestic fowl through ultrastructural, histochemical, biochemical and GLCM analysisPLOS ONE

Dear Dr. Kar,

Thank you for submitting your manuscript to PLOS ONE. After careful consideration, we feel that it has merit but does not fully meet PLOS ONE’s publication criteria as it currently stands. Therefore, we invite you to submit a revised version of the manuscript that addresses the points raised during the review process.

Please submit your revised manuscript by %DATE_REVISION _DUE%. If you will need more time than this to complete your revisions, please reply to this message or contact the journal office at plosone@plos.org. Please include the following items when submitting your revised manuscript:A rebuttal letter that responds to each point raised by the academic editor and reviewer(s). You should upload this letter as a separate file labeled 'Response to Reviewers'.A marked-up copy of your manuscript that highlights changes made to the original version. You should upload this as a separate file labeled 'Revised Manuscript with Track Changes'.An unmarked version of your revised paper without tracked changes. You should upload this as a separate file labeled 'Manuscript'.

We look forward to receiving your revised manuscript.

Kind regards,

Josué de Moraes, Ph.D.

Academic Editor

PLOS ONE

Journal Requirements:

2. Please note that PLOS ONE has specific guidelines on code sharing for submissions in which author-generated code underpins the findings in the manuscript. In these cases, all author-generated code must be made available without restrictions upon publication of the work. 

Please review our guidelines at https://journals.plos.org/plosone/s/materials-and-software-sharing#loc-sharing-code and ensure that your code is shared in a way that follows best practice and facilitates reproducibility and reuse.

3. Please provide the full species name and source for the worms used.

**Additional Editor Comments:**

Reviewers have now commented on your paper. You will see that they are advising that you revise your manuscript. If you are prepared to undertake the work required, I would be pleased to reconsider my decision. Moreover, the writing must be seriously improved. In several occasions phrases seems not have connection with the following phrase, besides lack of scientific accuracy in several parts of manuscript and typographic mistakes.

Reviewers' comments:

Reviewer's Responses to Questions

**Comments to the Author**

1. Is the manuscript technically sound, and do the data support the conclusions?

Reviewer #1: Yes

Reviewer #2: Yes

Reviewer #3: Partly

2. Has the statistical analysis been performed appropriately and rigorously? 

Reviewer #1: Yes

Reviewer #2: Yes

Reviewer #3: N/A

3. Have the authors made all data underlying the findings in their manuscript fully available?

Reviewer #1: Yes

Reviewer #2: Yes

Reviewer #3: Yes

4. Is the manuscript presented in an intelligible fashion and written in standard English?

Reviewer #1: Yes

Reviewer #2: Yes

Reviewer #3: No

5. Review Comments to the Author

Reviewer #1: Comments to the authors of the article:

1- It is recommended that in the introduction part of the article, a short explanation about the properties of chemical compounds and active ingredients of Holarrhena pubescens plant extract is given.

2- Raillietina worms isolated from the intestines of birds survived for how many hours in the culture media and laboratory conditions, and by what index is the boundary between the paralysis of the worms and the complete death of the worms determined?

3- It is better for the authors to use more recent references (after 2015) in the article. For example, it is recommended that from the article [Garedaghi yagoob, Khaki, A., Raza, S.H.A., (...), Abdelgayed, S.S., Kakar, M.U. Epidemiological and pathological studies on the helminthic parasites in native chickens of Tabriz city, Iran. Genetics and Molecular Research. 2017; 16(4),gmr16039824. DOI http://dx.doi.org/10.4238/gmr16039824. ] to use in them article references.

4- It is recommended that in the materials and methods section of the article, the exact geographical location (for example, the exact latitude and longitude) of the sampling location of the Holarrhena pubescens plant, which is in the region of West Bengal, India, and also the sampling year should be mentioned.

Reviewer #2: The manuscript “Anthelmintic efficacy of Holarrhena pubescens against Raillietina spp. in domestic fowl through ultrastructural, histochemical, biochemical and GLCM analysis” it is interesting and presents good execution in the methodology. However, I would like the authors to consider the following comments:

- In section: Introduction

The introduction is poor. I suggest describing biological, morphological characters, zoonotic and veterinary importance of Raillietin spp. once infections are highly pathogenic in domestic chickens (Gallus gallus domesticus), for example.

In addition, reporting the role of man as an accidental host in Raillietina spp infections. Human infection is rare, but has been reported from Central and South America, Cuba, Iran, Japan, Southeast Asia, the Philippines, French Polynesia, Australia, and the United States (Hawaii).

Please specify the real objective of this study. Why was Raillietina spp chosen as a study model?

-In section: Materials and Methods- Experiments- Efficacy Testing, line 86.

Write more details as parasites were exposed to extracts. Petri dish? Culture plates? medium of culture?

-In section: Results

Fig. 1. The control group cause Mortality of Raillietina spp. ???

Please insert subtitles on the Y axis of the graph (mortality in hours)

-In section: Discussion

The discussion is poor. I suggest discussing the importance of morphological changes in Raillietina spp. presented in SEM and MET, on the influence on the parasite- host relationship.

In addition, emphasize the need to develop new therapeutic alternatives in the veterinary parasitology field, since Raillietina spp. it represents such importance.

Reviewer #3: The reviewed manuscript describes the obtention and study of anthelmintic properties of ethanolic crude extract and their fractions obtained from different solvents from steam bark of Holarrhena pubescens against Raillietina spp. The manuscript affirms the efficacy of extracts base on comparison with Praziquantel (time of paralysis and time of death), besides observation of effects in parasites by Scanning Electron Microscopy, Transmission Electron Microscopy for instance.

The addition of chromatographic profile, dereplication or any other real information about extract and active fractions would improve the manuscript since it does not afford any information about real composition of evaluated extract and fractions.

The manuscript needs considerably improve in terms of scientific accuracy and correction of several typographic mistakes.

The manuscript in the present format is acceptable for publishing in Plos One since the recommendations below should be taken in consideration or answered.

Major corrections:

1. The authors must carefully and judiciously review the writing of manuscript by complete. There are several points that must be rewrite to text become clearer and present scientific accuracy that is needed.

Minor corrections:

1. It would be possible add consistent information of composition of extracts and active fractions of plant species, besides literature data cited?

2. Plant species can vary easily the chemical composition depending on period of year. Add the date of obtention of plant species.

3. Identification of plant species is important to attest its the real identity? Please afford the A the nunber of voucher specimen and Herbarium where it was deposited.

4. (Line17, 18): The authors say that manuscript focuses on ethanolic extract and ethyl acetate fraction, however fractions obtained from different solvents were evaluated in the efficacy test. As result, EtOH and AcOEt presented more efficacy and then this extracts were subject to following procedures. Please could you explain why n-BuOH was not selected for following tests, since the values are quite similar with EtOH extract (Table 1)?

5. (Line 70) Could you explain the observation “(without chlorophyll)”? What does this mean? Any method to remove chlorophyll was used or just to mention that this part of plant is free of chlorophyll?

6. (line 72) “The dried bark was processed…”. What exactly is processed? Please be clear. This phase as others presents in the manuscript lack of scientific accuracy. Probably, processed means “submitted to extraction”. However, the extraction method is described at lines 75, 76, 77 and 78. The authors must rewrite this section aiming a better description of complete process (since obtention of plant material to fraction obtention).

7. (line72) The extraction process took 15-20 days? Could you explain why and describes in detail the complete methodology?

8. (line 248) In discussion section authors bring to manuscript information about phytochemistry aspects of studied plant species base on literature. However, all these points raised by authors necessarily don’t reflect any considerably discussion to paper in this section. These points must be in introduction to present bioactive compounds of studied species. If authors had performed LC-MS/MS analysis of ETOH extract and/or active fractions and dereplication of compounds, a discussion could explored in this section.

9. (line 306) The section conclusion should be rewrite, since points mentioned don’t bring the idea of conclusion of presented work. The ideas should be expanded. I recommend the authors don’t use sentences as “The result of various experiments has shown”.

6. PLOS authors have the option to publish the peer review history of their article (what does this mean?). If published, this will include your full peer review and any attached files.

Reviewer #1: **Yes: **Yagoob Garedaghi

Department of Parasitology, Faculty of Veterinary Medicine, Tabriz Medical Sciences, Islamic Azad University, Tabriz, Iran.

Email: Yagoob.garedaghi@gmail.com

https://orcid.org/0000-0003-2976-2706

Reviewer #2: No

Reviewer #3: No

---

## [Author Response · Author response to Decision Letter 0]

12 Jul 2023

Dear Sir,

With respect to the comments made by the reviewers for our submitted article “Anthelmintic efficacy of Holarrhena pubescens against Raillietina spp. of domestic fowl through ultrastructural, histochemical, biochemical and GLCM analysis” I would like to submit our rebuttal as under.

Response:

All the style requirements of PLOS ONE including those for file naming have been met with in accordance to the links provided.

2. Please note that PLOS ONE has specific guidelines on code sharing for submissions in which author-generated code underpins the findings in the manuscript. In these cases, all author-generated code must be made available without restrictions upon publication of the work. 

Please review our guidelines at https://journals.plos.org/plosone/s/materials-and-software-sharing#loc-sharing-code and ensure that your code is shared in a way that follows best practice and facilitates reproducibility and reuse.

Response:

DOI 10.1101/2023.02.07.527541

3. Please provide the full species name and source for the worms used.

Response:

Two species of Raillietina occur mostly in the study area namely R.echinobothrida and R. tetragona with the prevalence of R.echinobothrida being higher according to our observations. However, since the sample worms obtained were not segregated according to species during the study, we preferred to use Raillietina spp. in denoting the model organism.

The worms were obtained after dissecting the intestines of freshly slaughtered local fowl (Gallus gallus domesticus) procured from the local market of Cooch Behar, West Bengal, India. We have mentioned the same in the Materials and Methods section, line 148-150.

Response:

All the minimal data set underlying the results have been uploaded as Supplementary Information files [S1_Table, S1_Fig, S4_Table and S8_Fig, S4_Table and S8_Fig(2)] in the submission portal. There are no ethical or legal restrictions to sharing our data.

Response:

Done via the online submission form.

 

Additional Editor Comments:

Reviewers have now commented on your paper. You will see that they are advising that you revise your manuscript. If you are prepared to undertake the work required, I would be pleased to reconsider my decision. Moreover, the writing must be seriously improved. In several occasions phrases seems not have connection with the following phrase, besides lack of scientific accuracy in several parts of manuscript and typographic mistakes.

Reviewers' comments:

Reviewer's Responses to Questions

Comments to the Author

1. Is the manuscript technically sound, and do the data support the conclusions?

Reviewer #1: Yes

Reviewer #2: Yes

Reviewer #3: Partly

2. Has the statistical analysis been performed appropriately and rigorously?

Reviewer #1: Yes

Reviewer #2: Yes

Reviewer #3: N/A

3. Have the authors made all data underlying the findings in their manuscript fully available?

Reviewer #1: Yes

Reviewer #2: Yes

Reviewer #3: Yes

4. Is the manuscript presented in an intelligible fashion and written in standard English?

Reviewer #1: Yes

Reviewer #2: Yes

Reviewer #3: No

 

5. Reviewer Comments to the Author

Reviewer #1: Comments to the authors of the article:

1- It is recommended that in the introduction part of the article, a short explanation about the properties of chemical compounds and active ingredients of Holarrhena pubescens plant extract is given.

Response:

We have included a short account of the chemical compounds and active ingredients of Holarrhena pubescens plant extracts from literature in the introduction section in two paragraphs from line 104-123 in the revised manuscript.

2- Raillietina worms isolated from the intestines of birds survived for how many hours in the culture media and laboratory conditions, and by what index is the boundary between the paralysis of the worms and the complete death of the worms determined?

Response:

The control worms survived for about 72 hours in the culture media and laboratory conditions. The boundary between paralysis and death of the worms were determined by the absence of movement in the paralysed worms when transferred to warm PBS. We have included the description in the Materials and methods section line no. 157-161 in the revised manuscript.

3- It is better for the authors to use more recent references (after 2015) in the article. For example, it is recommended that from the article [Garedaghiyagoob, Khaki, A., Raza, S.H.A., (...), Abdelgayed, S.S., Kakar, M.U. Epidemiological and pathological studies on the helminthic parasites in native chickens of Tabriz city, Iran. Genetics and Molecular Research. 2017; 16(4),gmr16039824. DOI http://dx.doi.org/10.4238/gmr16039824. ] to use in them article references.

Response:

As advised, we have included new and recent references in our manuscript. The number of references have increased from 40 to 68 in the revised manuscript.

4- It is recommended that in the materials and methods section of the article, the exact geographical location (for example, the exact latitude and longitude) of the sampling location of the Holarrhenapubescens plant, which is in the region of West Bengal, India, and also the sampling year should be mentioned.

Response:

Fresh bark was collected from the plant H. pubescens from the town of Cooch Behar (26.3169° N, 89.4457° E), West Bengal, India in month of April- June, Since 2018 till date. We have added it in the Materials and method section line no. 134-135 in the revised manuscript.

Reviewer #2: The manuscript “Anthelmintic efficacy of Holarrhena pubescens against Raillietina spp. in domestic fowl through ultrastructural, histochemical, biochemical and GLCM analysis” it is interesting and presents good execution in the methodology. However, I would like the authors to consider the following comments:

- In section: Introduction

The introduction is poor. I suggest describing biological, morphological characters, zoonotic and veterinary importance of Raillietin spp. once infections are highly pathogenic in domestic chickens (Gallus gallusdomesticus), for example.

In addition, reporting the role of man as an accidental host in Raillietinaspp infections. Human infection is rare, but has been reported from Central and South America, Cuba, Iran, Japan, Southeast Asia, the Philippines, French Polynesia, Australia, and the United States (Hawaii).

Please specify the real objective of this study. Why was Raillietina spp. chosen as a study model?

Response:

The introduction section has been restructured considering the comments of the reviewer and all the mentioned points have been included in the revised manuscript.

-In section: Materials and Methods- Experiments- Efficacy Testing, line 86.

Write more details as parasites were exposed to extracts. Petri dish? Culture plates? medium of culture?

Response:

The parasites were cultured in Phosphate Buffer Saline (PBS), pH 7.4 at 370C in petri dishes in an incubator. We have added this in line no. 154 in Materials and Methods section in the revised manuscript.

-In section: Results

Fig. 1. The control group cause Mortality of Raillietina spp. ???

Response:

No, but the worms survived in PBS for a limited period of time i.e.about 71 hours.

Please insert subtitles on the Y axis of the graph (mortality in hours)

Response:

Done as advised.

-In section: Discussion

The discussion is poor. I suggest discussing the importance of morphological changes in Raillietina spp. presented in SEM and MET, on the influence on the parasite- host relationship.

In addition, emphasize the need to develop new therapeutic alternatives in the veterinary parasitology field, since Raillietina spp. it represents such importance.

Response:

The paragraph in the Discussion section line no. 329- 341 has been rewritten to include the points mentioned by the reviewer. Other sections have been rephrased as well as new information has been added to augment the discussion. 

 

Reviewer #3: The reviewed manuscript describes the obtention and study of anthelmintic properties of ethanolic crude extract and their fractions obtained from different solvents from steam bark of Holarrhena pubescens against Raillietina spp. The manuscript affirms the efficacy of extracts base on comparison with Praziquantel (time of paralysis and time of death), besides observation of effects in parasites by Scanning Electron Microscopy, Transmission Electron Microscopy for instance.

The addition of chromatographic profile, dereplication or any other real information about extract and active fractions would improve the manuscript since it does not afford any information about real composition of evaluated extract and fractions.

The manuscript needs considerably improve in terms of scientific accuracy and correction of several typographic mistakes.

The manuscript in the present format is acceptable for publishing in Plos One since the recommendations below should be taken in consideration or answered.

Major corrections:

1. The authors must carefully and judiciously review the writing of manuscript by complete. There are several points that must be rewrite to text become clearer and present scientific accuracy that is needed.

Response:

Many parts of the manuscript have been rephrased as well as additional points have been included as suggested to improve the quality of English as well as clarity of scientific expression in the revised manuscript.

Minor corrections:

1. It would be possible add consistent information of composition of extracts and active fractions of plant species, besides literature data cited?

Response:

Information about the composition of the extracts have been added in the introduction section line no. 104-123 in the revised manuscript from literature. However, data on the phytochemical characterisation of our plant extract is still not presentable though we plan to publish the same in the near future.

2. Plant species can vary easily the chemical composition depending on period of year. Add the date of obtention of plant species.

Response:

2018, added in Materials and Methods section line no. 134-135 of the revised manuscript.

3. Identification of plant species is important to attest it’s the real identity? Please afford the A the nunber of voucher specimen and Herbarium where it was deposited.

Respose.

Dr. Monoranjan Chowdhury identified the plant and the accession number is- 12638. It was kept at herbarium of Department of Botany, University of North Bengal, Darjeeling, West Bengal. Mention in Materials and Methods section line no. 135-137 of the revised manuscript

4. (Line17, 18): The authors say that manuscript focuses on ethanolic extract and ethyl acetate fraction, however fractions obtained from different solvents were evaluated in the efficacy test. As result, EtOH and AcOEt presented more efficacy and then this extracts were subject to following procedures. Please could you explain why n-BuOH was not selected for following tests, since the values are quite similar with EtOH extract (Table 1)?

Response:

The ethanolic extract was first used for efficacy testing. Then the same extract was then subjected to fractionation using hexane, chloroform, ethyl acetate and n-butanol. These four fractions obtained from the ethanolic extract were again used for efficacy testing, out of which the ethyl acetate fraction showed the highest efficacy.

5. (Line 70) Could you explain the observation “(without chlorophyll)”? What does this mean? Any method to remove chlorophyll was used or just to mention that this part of plant is free of chlorophyll?

Response:

No method of removing chlorophyll was used. However, the outer layer of the obtained bark was peeled to obtain the fresh brownish inner layer. Nevertheless, we have removed the phrase “without chlorophyll” from the revised manuscript.

6. (line 72) “The dried bark was processed…”. What exactly is processed? Please be clear. This phase as others presents in the manuscript lack of scientific accuracy. Probably, processed means “submitted to extraction”. However, the extraction method is described at lines 75, 76, 77 and 78. The authors must rewrite this section aiming a better description of complete process (since obtention of plant material to fraction obtention).

Response:

We have rephrased the mentioned section in the Materials and Methods line no. 135-145 in the revised manuscript to improve the scientific accuracy.

7. (line72) The extraction process took 15-20 days? Could you explain why and describes in detail the complete methodology?

Response:

The dried bark was kept immersed in ethanol for 15-20 days with repeated stirring to maximize the yield of phytochemicals. 

8. (line 248) In discussion section authors bring to manuscript information about phytochemistry aspects of studied plant species base on literature. However, all these points raised by authors necessarily don’t reflect any considerably discussion to paper in this section. These points must be in introduction to present bioactive compounds of studied species. If authors had performed LC-MS/MS analysis of ETOH extract and/or active fractions and dereplication of compounds, a discussion could explored in this section.

Response:

As advised, we have moved the section dealing with the phytochemistry aspects of the plant to the Introduction section line no. 104-123 in the revised manuscript.

We have not performed any LC-MS/MS analysis of the extract yet.

9. (line 306) The section conclusion should be rewrite, since points mentioned don’t bring the idea of conclusion of presented work. The ideas should be expanded. I recommend the authors don’t use sentences as “The result of various experiments has shown”.

Response:

The conclusion has been rewritten as advised.

6. PLOS authors have the option to publish the peer review history of their article (what does this mean?). If published, this will include your full peer review and any attached files.

We do not wish to make the peer review history public.

Hope this will suffice for now. We are open for discussion for any further comments made by the reviewers.

Thanking you,

Yours sincerely,

Dr. Pradip Kumar Kar

---

## [Decision Letter · Decision Letter 1]

26 Jul 2023

PONE-D-23-03097R1Anthelmintic efficacy of * Holarrhena pubescens * against * Raillietina * spp. of domestic fowl through ultrastructural, histochemical, biochemical and GLCM analysisPLOS ONE

Dear Dr. Kar,

Thank you for submitting your manuscript to PLOS ONE. After careful consideration, we feel that it has merit but does not fully meet PLOS ONE’s publication criteria as it currently stands. Therefore, we invite you to submit a revised version of the manuscript that addresses the points raised during the review process. The authors have made significant improvements to the manuscript compared to the first version. However, there are still some typographical errors and sentences that need to be rephrased to enhance the text's conciseness and fluency, which are crucial for a scientific article. I recommend that the authors thoroughly revise the text to enhance its English quality. Utilizing the services of a native English speaker would be beneficial in this regard.

We look forward to receiving your revised manuscript.

Kind regards,

Josué de Moraes, Ph.D.

Academic Editor

PLOS ONE

Additional Editor Comments (if provided):

I recommend that the authors thoroughly revise the text to enhance its English quality. Utilizing the services of a native English speaker would be beneficial in this regard.

Reviewers' comments:

Reviewer's Responses to Questions

**Comments to the Author**

1. If the authors have adequately addressed your comments raised in a previous round of review and you feel that this manuscript is now acceptable for publication, you may indicate that here to bypass the “Comments to the Author” section, enter your conflict of interest statement in the “Confidential to Editor” section, and submit your "Accept" recommendation.

Reviewer #1: All comments have been addressed

Reviewer #3: (No Response)

2. Is the manuscript technically sound, and do the data support the conclusions?

Reviewer #1: Yes

Reviewer #3: Yes

3. Has the statistical analysis been performed appropriately and rigorously? 

Reviewer #1: I Don't Know

Reviewer #3: Yes

4. Have the authors made all data underlying the findings in their manuscript fully available?

Reviewer #1: Yes

Reviewer #3: Yes

5. Is the manuscript presented in an intelligible fashion and written in standard English?

Reviewer #1: Yes

Reviewer #3: Yes

6. Review Comments to the Author

Reviewer #1: it is recommended that from the article [Garedaghi yagoob, Khaki, A., Raza, S.H.A., (...), Abdelgayed, S.S., Kakar, M.U. Epidemiological and pathological studies on the helminthic parasites in native chickens of Tabriz city, Iran. Genetics and Molecular Research. 2017; 16(4),gmr16039824. DOI http://dx.doi.org/10.4238/gmr16039824. ] to use in them article references.

Reviewer #3: The authors improved the manuscript compared to the first version. However, it is necessary several adjustments in the text taking into consideration typographic mistakes and phrases that must be rewritten to let the text concise and fluid which is necessary for a scientific paper. Unfortunately, the section that describes the extraction procedure, as well as the obtention of organic phases, remains without details. Even though the focus of the work is the evaluation of anthelmintic properties of extract and organic phases of H. pubescens, details of the obtention of these materials are required and essential. As a final consideration, it would be helpful to submit the text to a scientific-technical translation company to adjust the manuscript to be published.

Bellow follows specific comments and considerations to the authors.

Minor corrections:

1. Please, maintain a solvent nomenclature standardized. Use the first letter capitalized for all of them or none of them (except at the beginning of a phrase, for instance);

2. Use italics for “n” before “-butanol”.

3. Please, pay attention to spaces between characters, punctuation, and hyphen. Several points must be reviewed.

4. The same aspect is seen for units after numbers. Please follow a pattern and check spaces between numbers and units.

5. Lines 143 and 144 – Unfortunately the procedure described is not clear enough. Please be specific as much as you can.

6. Lines 143 and 144 – It is necessary to inform the right procedure of liquid-liquid extraction as requested previously. Be specific in terms of the used mass of crude extract, how this crude extract was solubilized (what solvent), or partially solubilized, the addition order of solvents, quantities of used solvents the final mass of each phase obtained, for instance.

7. Is not completely clear how and which procedures (liquid-liquid or fractional distillation) were employed or in which stage each of them was used.

8. Line 144 – Could you explain how fractional distillation was employed to obtain the organic phases?.

9. Despite information about the chemistry of plant species was included in the introduction, the text must be reviewed from lines 100 to 122 to become more concise and let the text fluid. The whole idea of this part is impaired by the repetition of ideas and phrases that should appear in a better position to bring consistency to the text.

10. Line 111 – Please, replace “Kutaja” with the scientific name of the plant species.

11. Lines 114 to 117 – Please, include the reference about alkaloids in H. pubescens.

12. From lines 134 to 137 – This part of the text needs to be rewritten to bring fluidity.

13. In many parts of the manuscript, the authors don’t let clear the amount of material or time of procedure used. For example, lines 137, 138, 141, and other parts. Why is not shown the exact amount of material and time used?.

14. Line 141 – As a suggestion replace “stored in a container at” with “stored under refrigeration at”.

15. Line 155 mentions a 20 mg/mL concentration in the row of samples tested for efficacy testing. However, it is not seen this concentration in Table 1. Please add the value in the table or remove the information (line 155).

7. PLOS authors have the option to publish the peer review history of their article (what does this mean?). If published, this will include your full peer review and any attached files.

Reviewer #1: **Yes: **Yagoob Garedaghi

1. Department of Parasitology, Faculty of Veterinary Medicine, Tabriz Medical Sciences, Islamic Azad University, Tabriz, Iran.

https://orcid.org/0000-0003-2976-2706

Reviewer #3: No

---

## [Author Response · Author response to Decision Letter 1]

1 Aug 2023

Dear Sir,

With respect to the comments made by the reviewers for our submitted article “Anthelmintic efficacy of Holarrhena pubescens against Raillietina spp. Of domestic fowl through ultrastructural, histochemical, biochemical and GLCM analysis” I would like to submit our rebuttal as under.

Additional Editor Comments (if provided):

I recommend that the authors thoroughly revise the text to enhance its English quality. Utilizing the services of a native English speaker would be beneficial in this regard.

Response

Response:

The authors revised the manuscript as and where necessary, and, would like to state that the quality of English of this communication conforms to existing standard comparable to articles published in reputed high-impact journals. The author would like to further reiterate that this manuscript has been written as per convention practiced by Parasitologists, world over, for publication of anthelmintic efficacy data. The corresponding author has already published two articles in Plos One which has been appreciated by the scientific community.

1) Pradip Kumar Kar, Sanatan Murmu, Saswati Saha, Veena Tandon, Krishnendu Acharya. Anthelmintic efficacy of gold nanoparticles derived from phytopathogenic fungi Nigrospora oryzae. Plos One (2014) 9(1): e84693, ISSN No. – 1932-6203, January 2014

2) Anirban Ash, Tomáš Scholz, Alain de Chambrier, Jan Brabec, Mikuláš Oros, Pradip Kumar Kar, Shivaji Prabhakar Chavan, Jean Mariaux. Revision of Gangesia (Cestoda: Proteocephalidea) in the Indomalayan Region: morphology, molecules and surface ultrastructure. Plos One (2012) 7(10): 1-28, e46421 ISSN No. – 1932-6203, October 2012

Hence, the authors would be highly obliged if this explanation suffices the quandary on the quality of English of this article, and would like to provide further clarification / justification if so desired by the editorial team.

Reviewers' comments:

Reviewer's Responses to Questions

Comments to the Author

1. If the authors have adequately addressed your comments raised in a previous round of review and you feel that this manuscript is now acceptable for publication, you may indicate that here to bypass the “Comments to the Author” section, enter your conflict of interest statement in the “Confidential to Editor” section, and submit your "Accept" recommendation.

Reviewer #1: All comments have been addressed

Reviewer #3: (No Response)

2. Is the manuscript technically sound, and do the data support the conclusions?

Reviewer #1: Yes

Reviewer #3: Yes

3. Has the statistical analysis been performed appropriately and rigorously?

Reviewer #1: I Don't Know

Reviewer #3: Yes

4. Have the authors made all data underlying the findings in their manuscript fully available?

Reviewer #1: Yes

Reviewer #3: Yes

5. Is the manuscript presented in an intelligible fashion and written in standard English?

Reviewer #1: Yes

Reviewer #3: Yes

6. Review Comments to the Author

Reviewer #1: it is recommended that from the article [Garedaghiyagoob, Khaki, A., Raza, S.H.A., (...), Abdelgayed, S.S., Kakar, M.U. Epidemiological and pathological studies on the helminthic parasites in native chickens of Tabriz city, Iran. Genetics and Molecular Research. 2017; 16(4),gmr16039824. DOI http://dx.doi.org/10.4238/gmr16039824. ] to use in them article references.

Response:

It has been done. Line number: 72

Reviewer #3: The authors improved the manuscript compared to the first version. However, it is necessary several adjustments in the text taking into consideration typographic mistakes and phrases that must be rewritten to let the text concise and fluid which is necessary for a scientific paper. Unfortunately, the section that describes the extraction procedure, as well as the obtention of organic phases, remains without details. Even though the focus of the work is the evaluation of anthelmintic properties of extract and organic phases of H. pubescens, details of the obtention of these materials are required and essential. As a final consideration, it would be helpful to submit the text to a scientific-technical translation company to adjust the manuscript to be published.

Bellow follows specific comments and considerations to the authors.

Minor corrections:

1. Please, maintain a solvent nomenclature standardized. Use the first letter capitalized for all of them or none of them (except at the beginning of a phrase, for instance);

Response:

Done as advised.

2. Use italics for “n” before “-butanol”.

Response:

Done as advised.

3. Please, pay attention to spaces between characters, punctuation, and hyphen. Several points must be reviewed.

Response:

Done as advised.

4. The same aspect is seen for units after numbers. Please follow a pattern and check spaces between numbers and units.

Response:

Done as advised.

5. Lines 143 and 144 – Unfortunately the procedure described is not clear enough. Please be specific as much as you can.

Response:

50 ml of aqueous crude extract solution (2.5 grm in 50 ml distilled water) was combined with 50 ml of Hexane and left to stand for 2 hours. The Hexane fraction was subsequently removed from the upper phase using the liquid liquid separation technique. The method was repeated three times using Chloroform (lower phase), Ethyl acetate, and n-Butanol. The final volume of each fraction (150 ml) was collected and dried using a Rotary Vacuum Evaporator (Buchi Rotavapor R-100). Around 0.25 g, 0.14 g, 0.92 g and 1.1 g of Hexane, Chloroform, Ethyl acetate and n-Butanol fractions were obtained from 2.5 g of crude extract respectively. Mentioned in Materials and Methods section line no. 143-148 of the revised manuscript

6. Lines 143 and 144 – It is necessary to inform the right procedure of liquid-liquid extraction as requested previously. Be specific in terms of the used mass of crude extract, how this crude extract was solubilized (what solvent), or partially solubilized, the addition order of solvents, quantities of used solvents the final mass of each phase obtained, for instance.

Response:

Mentioned in Materials and Methods section line no. 143-148 of the revised manuscript

7. Is not completely clear how and which procedures (liquid-liquid or fractional distillation) were employed or in which stage each of them was used.

Response:

Mentioned in Materials and Methods section line no. 143-148 of the revised manuscript

8. Line 144 – Could you explain how fractional distillation was employed to obtain the organic phases?.

Response:

Mentioned in Materials and Methods section line no. 143-148 of the revised manuscript

9. Despite information about the chemistry of plant species was included in the introduction, the text must be reviewed from lines 100 to 122 to become more concise and let the text fluid. The whole idea of this part is impaired by the repetition of ideas and phrases that should appear in a better position to bring consistency to the text.

Response:

Done as advised.

10. Line 111 – Please, replace “Kutaja” with the scientific name of the plant species.

Response:

Done as advised.

11. Lines 114 to 117 – Please, include the reference about alkaloids in H. pubescens.

Response:

Included in line number 115.

12. From lines 134 to 137 – This part of the text needs to be rewritten to bring fluidity.

Response:

Done as advised.

13. In many parts of the manuscript, the authors don’t let clear the amount of material or time of procedure used. For example, lines 137, 138, 141, and other parts. Why is not shown the exact amount of material and time used?.

Response:

The plant materials were finely chopped and air dried for 24-36 hours after being collected. The dried bark components were soaked in ethanol in a glass container (100 grams in 500 ml) for 15-20 days with frequent stirring. The solution was then filtered using Whatman Filter Paper (No. 14), and then dried using a Rotary Vacuum Evaporator (Buchi Rotavapor R-100). After drying, the collected crude extracts of plant material were stored under refrigeration at 4°C until further use. Around 7.0-8.0 gram (g) of plant extract was obtained from 100 g of plant material soaked in 500 millilitre (ml) ethanol.

It has been clearly mentioned in the main manuscript earlier, line number 135 -142 that only stem bark is used for crude extract preparation and other subsequent experiments.

14. Line 141 – As a suggestion replace “stored in a container at” with “stored under refrigeration at”.

Response:

Done as advised.

15. Line 155 mentions a 20 mg/mL concentration in the row of samples tested for efficacy testing. However, it is not seen this concentration in Table 1. Please add the value in the table or remove the information (line 155).

Response:

Removed the information.

7. PLOS authors have the option to publish the peer review history of their article (what does this mean?). If published, this will include your full peer review and any attached files.

Response:

We do not wish to make the peer review history public.

Hope this will suffice for now. We are open for discussion for any further comments made by the reviewers.

Thanking you,

Yours sincerely,

Dr. Pradip Kumar Kar

---

## [Decision Letter · Decision Letter 2]

21 Aug 2023

Anthelmintic efficacy of * Holarrhena pubescens * against * Raillietina * spp. of domestic fowl through ultrastructural, histochemical, biochemical and GLCM analysis

PONE-D-23-03097R2

Dear Dr. Kumar Kar,

We’re pleased to inform you that your manuscript has been judged scientifically suitable for publication and will be formally accepted for publication once it meets all outstanding technical requirements.

Kind regards,

Josué de Moraes, Ph.D.

Academic Editor

PLOS ONE

Additional Editor Comments (optional):

Reviewers' comments:

Reviewer's Responses to Questions

**Comments to the Author**

1. If the authors have adequately addressed your comments raised in a previous round of review and you feel that this manuscript is now acceptable for publication, you may indicate that here to bypass the “Comments to the Author” section, enter your conflict of interest statement in the “Confidential to Editor” section, and submit your "Accept" recommendation.

Reviewer #3: All comments have been addressed

2. Is the manuscript technically sound, and do the data support the conclusions?

Reviewer #3: Yes

3. Has the statistical analysis been performed appropriately and rigorously? 

Reviewer #3: N/A

4. Have the authors made all data underlying the findings in their manuscript fully available?

Reviewer #3: Yes

5. Is the manuscript presented in an intelligible fashion and written in standard English?

Reviewer #3: Yes

6. Review Comments to the Author

Reviewer #3: (No Response)

7. PLOS authors have the option to publish the peer review history of their article (what does this mean?). If published, this will include your full peer review and any attached files.

Reviewer #3: No

---

## [Editor Report · Acceptance letter]

1 Sep 2023

PONE-D-23-03097R2 

Anthelmintic efficacy of *Holarrhena pubescens* against *Raillietina* spp. of domestic fowl through ultrastructural, histochemical, biochemical and GLCM analysis 

Dear Dr. Kar:

I'm pleased to inform you that your manuscript has been deemed suitable for publication in PLOS ONE. Congratulations! Your manuscript is now with our production department. 

Kind regards, 

on behalf of

Dr. Josué de Moraes 

Academic Editor

PLOS ONE